# Mobile App-Based Coaching for Alcohol Prevention among Adolescents: Pre–Post Study on the Acceptance and Effectiveness of the Program “MobileCoach Alcohol”

**DOI:** 10.3390/ijerph20043263

**Published:** 2023-02-13

**Authors:** Severin Haug, Nikolaos Boumparis, Andreas Wenger, Raquel Paz Castro, Michael Patrick Schaub

**Affiliations:** 1Swiss Research Institute for Public Health and Addiction, Zurich University, Konradstrasse 32, 8005 Zurich, Switzerland; 2Marie Meierhofer Childrens’s Institute, Pfingstweidstrasse 16, 8005 Zurich, Switzerland

**Keywords:** alcohol, prevention, students, adolescents, mobile app

## Abstract

Background: At-risk alcohol use, particularly binge drinking, is widespread among adolescents and young adults in most Western countries. *MobileCoach Alcohol* is a mobile app-based program for alcohol prevention that provides individualized coaching using a conversational agent. The current study tested the acceptance, use, and evaluation of this newly developed program and explored its potential effectiveness. Methods: Longitudinal pre–post study among upper secondary and vocational school students in Switzerland. Within the *MobileCoach Alcohol* prevention program, a virtual coach motivated participants to deal with alcohol sensitively, and provided feedback on alcohol use and strategies to resist alcohol for a period of 10 weeks. Information was provided in weekly dialogs, within contests with other participants, and interactive challenges. By means of a follow-up survey after the end of the 10-week program, indicators of the use, acceptance, and effectiveness of the program were examined. Results: Between October 2020 and July 2022, the program was advertised in upper secondary and vocational schools. Recruiting schools and school classes was difficult due to the COVID-19 containment measures in place during this period. Nevertheless, the program could be implemented in 61 upper secondary and vocational school classes with a total of 954 participating students. Three out of four students who were present in the school classes participated in the *MobileCoach Alcohol* program and the associated study. Online follow up assessment at week 10 was completed by 272 program participants (28.4%). Based on program use and evaluations by the participants, the overall acceptance of the intervention was good. The proportion of students who engaged in binge drinking was significantly reduced from 32.7% at baseline to 24.3% at follow up. Furthermore, the longitudinal analyses revealed decreases in the maximum number of alcoholic drinks consumed on an occasion and the mean number of standard drinks per month, whereas self-efficacy to resist alcohol increased between baseline and follow up. Conclusions: The mobile app-based *MobileCoach Alcohol* program proved to be an attractive intervention, in which the majority of students were interested when proactively recruited at school classes. It allows for individualized coaching in large groups of adolescents and young adults and is promising for reducing at-risk alcohol use.

## 1. Introduction

Addictive behaviors, such as the use of alcohol, nicotine, and cannabis, or the excessive use of online media, are widespread in late adolescence and early adulthood. The results of a representative survey in 2021 among 15- to 19-year-olds in Switzerland show regular, i.e., at least monthly, use of cigarettes by 16%, e-cigarettes by 6%, cannabis by 9%, and alcohol by 52% of respondents [1].

At-risk alcohol use among adolescents and young adults is associated with social and educational problems, accidents, damage to the liver [2], and, in the longer term, increased risk of chronic diseases, such as heart and liver disease or alcohol dependence [3].

An important indicator of at-risk alcohol use among adolescents is binge drinking, which is internationally mostly defined as the consumption of four or more (for women) or five or more (for men) alcoholic beverages on one occasion [4]. Binge drinking is also common among adolescents and young adults from Switzerland. The results from the Swiss Health Survey 2017 show that 64% of men aged 15 to 25 years and 58% of women in this age group engaged in binge drinking within the past 12 months; at least monthly binge drinking was present in 30% of men and 24% of women [5].

Interventions for reducing at-risk alcohol use among adolescents and young adults reviewed in international studies are predominantly based on motivational interviewing or normative feedback [6]. However, the available Cochrane reviews conclude that the effectiveness of both approaches has been insufficiently demonstrated [7,8]. Overall, measures to teach protective behavioral strategies and interventions mediated via new communication technologies currently seem to be the most promising [3].

Because of the possibility of automated individualization of content as well as the strong use and high attractiveness of new communication technologies among adolescents and young adults, these technologies open up a promising route for conveying preventive information and counseling services [9]. In addition to Internet-based services, SMS- or smartphone app-based services in particular offer the possibility of motivating adolescents and adults to change their behavior by means of individualized information and to support them in doing so. A review on mobile-phone-based interventions to reduce alcohol use in young people concluded that these programs provide an acceptable, affordable, and potentially effective way for delivering information about reducing alcohol use [10]. From the included 18 studies, 9 reported a reduction in alcohol consumption; however, further methodologically sound studies are required to draw clearer conclusions about their effectiveness.

The mobile-phone-based program *MobileCoach Alcohol* for the reduction in at-risk alcohol use in adolescents was implemented in Swiss upper secondary and vocational schools in 2012 [11]. The web- and text-messaging-based program version included online feedback and individually tailored text messages addressing social norms, outcome expectations, motivation, self-efficacy, and planning processes, provided over 3 months. Between 2015 and 2017, the program was tested within a cluster randomized controlled trials in vocational and upper secondary school students in Switzerland [12,13]. The results of this trial revealed high acceptance and efficacy of the program among the target group of young people. Three of four students who were asked to take part in the program and the associated study accepted the invitation. The efficacy data showed a significant positive intervention effect on the main outcome, namely prevalence of binge drinking, which decreased by 5.9% in the intervention group and increased by 2.6% in the control group, relative to that of the baseline assessment.

While phone calls and communication via SMS were the most frequently used applications on cell phones until the early 2010s, they have been increasingly replaced by messenger-based communication via smartphone apps in recent years. Today, communication via messenger services (especially WhatsApp) is the most frequently used cell phone function among teenagers in Switzerland [14].

In parallel with the spread of smartphones, the question of whether to offer an app or SMS program to support health behavior change has been raised by several authors [15,16,17]. Both media have specific advantages and disadvantages, both for reaching program participants and for delivering effective interventions. Apps allow for a direct integration of media objects (e.g., images, photos, and video clips) and web links as graphic icons that are highly visible on the screen and can be easily accessed, shared, or saved within the app. In addition, apps are able to provide a richer user experience, but require users to proactively engage with the program, whereas SMSs allow for a more proactive approach, where users must actively choose not to engage with it.

Based on the potential advantages of mobile app-based programs and the changing digital communication behavior among young people, the *MobileCoach Alcohol* program was updated, optimized, and provided as mobile app-based program including coaching by a conversational agent. Within the current study, we (1) tested the acceptance, use, and evaluation of this newly developed app-based program version and (2) explored its potential effectiveness within a longitudinal pre–post study among vocational and upper secondary school students in Switzerland.

## 2. Methods

### 2.1. Participants, Setting, and Procedure

Vocational and upper secondary schools in the Swiss cantons of Aargau, Bern, and Zurich were invited for participation via e-mail by prevention experts from collaborating regional centers for addiction prevention. Interested teachers at the individual schools were informed about the objectives of the program. Furthermore, its implementation and procedure in the school classes were discussed. The teachers reserved a period of 20–30 min within the regular school lessons for the program implementation.

During the school lesson, reserved for program implementation, the students were informed about the program and the accompanying study by the prevention expert and invited to participate with the help of an introductory video available at www.mobile-coach.ch (accessed on 10 February 2023). Students were asked to download the app on their smartphones and to complete online baseline assessment and study registration. The inclusion criteria for this study were a minimum age of 15 years and ownership of a smartphone. After being screened for the inclusion criteria and giving informed consent, study participants were invited to choose a username. Subsequently, they received coaching by the conversational agent for a period of 10 weeks (see intervention program). All participants were invited within the app to complete an online follow-up assessment after program completion, 10 weeks after their enrollment in the study.

The Ethics Committee of the Faculty of Philosophy at the University of Zurich, Switzerland approved the study protocol (date of approval: 24 June 2020, approval no. 20.6.14).

### 2.2. Intervention Program

*MobileCoach Alcohol* (www.mobile-coach.ch, accessed on 10 February 2023) is a mobile app-based program for alcohol prevention that provides individualized coaching by a conversational agent. The native app was developed for both iOS and Android and is available via the respective stores in Switzerland. A web-based administration program enables an overview of all realized coaching dialogs and messages, the administration of the participants’ data, and the answering of the participants’ personal questions during the program implementation.

After choosing their language, participants could select a male or female coach to interact with. Subsequently, demographic variables (age and sex) and alcohol use were assessed in dialogue form by the coach.

All participants from a minimum age of 15 years were then informed about the program within the app and invited to participate. Program participants were then asked for a username and invited to fill in some items on their drinking refusal self-efficacy. Sections of the registration within the app are shown in Figure 1.

The intervention content was customized using data from the baseline assessment and program participants received individualized messages by the coach for a total of 10 weeks. An average weekly dialogue took between two and five minutes to process. The coach regularly contacted the participants via push notifications and guided them through the program. The coaching content was individualized based on age, sex, and alcohol consumption (at-risk or not at risk, days of consumption, times of consumption, and consumption situations). The coaching was intended to (1) stimulate positive outcome expectations regarding low-risk drinking, (2) increase self-efficacy to resist alcohol consumption, and (3) stimulate planning processes to drink no or less alcohol in typical drinking situations. For participants who practiced binge drinking, the day of the week and time of day with the highest alcohol consumption were assessed in order to send push messages shortly before drinking situations, e.g., Saturday evening at 21:00 p.m., in order to promote low-risk consumption.

The intervention elements of the coaching were based on the principles of social cognitive theory (e.g., self-monitoring and goal-setting) [18], the social norms approach (e.g., normative feedback) [19], and motivational interviewing (e.g., decisional balance) [20]. An overview of the intervention elements is shown in Table 1.

Interactive elements such as quizzes, challenges, and contests were designed to encourage cognitive engagement with the topic in a playful way. The contests were intended to establish a social connection with other program participants, and participants could upload pictures and messages on a beautiful experience without alcohol. These contributions were presented anonymously to the other program participants and could be marked with the help of a “like button”. The best contest contributions were then presented to all participants by the coach during a separate coaching session.

By actively interacting with the coach, participants could collect credits. The more credits they had, the higher the chances of winning one of several attractive prizes (10 × 50 Swiss Francs, 1 × 500 Swiss Francs), which were raffled off among all participants at the end of the program. Participants collected credits by completing the weekly dialogs with the coach, as well as by participating in the follow-up assessment. Screenshots of the intervention program are shown in Figure 2.

In addition to the weekly dialogs, which were initiated by the coach by a push notification, participants had the opportunity to initiate and process dialogs on several topics themselves, e.g., on myths about alcohol, on the long-term consequences of alcohol consumption, or on how to avoid a hangover.

Beyond the automated dialogs with the conversational agent, the program offered the option of posing individual questions on the subject of alcohol to an expert (Ask-the-Expert). Program users could pose their individual questions via chat within the app for a period of 3 weeks and received an answer within a few days from a prevention expert from the Swiss Research Institute for Public Health and Addiction.

### 2.3. Assessments and Outcomes

At baseline, we assessed the following individual demographic information: (1) age, (2) sex, and (3) migration background by assessing the country of birth of both parents. Additionally, characteristics of the schools and classes were collected. Both baseline and follow-up assessments included the subsequent alcohol-related measures:Binge drinking in the preceding 30 days, assessed by asking participants to report the number of standard drinks (10–12 g of pure alcohol) consumed on the heaviest drinking occasion in the preceding 30 days. Binge drinking was defined as drinking at least five drinks on a single occasion in men and four drinks on a single occasion in women.Maximum number of alcoholic standard drinks consumed on one occasion in the preceding 30 days.Total number of alcoholic standard drinks consumed in the previous 30 days by multiplying the number of alcohol consumption days in the preceding 30 days by the number of alcoholic standard drinks consumed on a typical drinking day.Self-efficacy to resist drinking alcohol in different situations ((a) when going out, (b) when someone offers you alcohol, and (c) when friends drink alcohol) based on items from the Drinking Refusal Self-Efficacy Questionnaire [21].

At follow-up, we additionally evaluated the acceptance of the program by asking the participants whether (1) they would participate in the program again, (2) whether it was fun to participate in the program, and (3) whether they would recommend the program to friends. Additionally, they should indicate whether the tips and information provided were (1) comprehensible, (2) helpful, and (3) relevant to their individual situation. Furthermore, program participants were asked to rate the program and different program elements using the response categories of very good, good, acceptable, and poor.

Beyond participants’ self-reported data, program use, operationalized by the number of completed weekly dialogs with the coach, was extracted from the log files of the *MobileCoach Alcohol* database.

### 2.4. Statistical Analysis

To test for baseline differences between program participants and non-participants, Pearson χ^2^ tests for dichotomous variables and Student’s t tests for metric variables were applied. For the attrition analysis (program participants lost to follow-up), we also used χ^2^ tests for categorical variables and t tests for continuous variables.

To assess the longitudinal course of the outcomes over the study period of 10 weeks, we used (generalized) linear mixed models ((G)LMMs). (G)LMMs take into account the dependence between observations caused by the clustering of data by participants, and allow for the number of observations to vary between participants [22]. We used generalized linear mixed models to assess intervention effects for binary and count outcomes, while linear mixed models (LMM) were used for continuous outcomes utilizing the lme4 package [22]. Within each (G)LMM, a random intercept was modelled for school class. Independent variables included time as a predictor. To control for attrition bias, we additionally added the respective baseline variables and variables on program use as covariates to the (G)LMMs. A type I error rate of *p* < 0.05 on two-sided tests was considered statistically significant. All of the analyses were performed using R, version 4.1.2 (R Foundation for Statistical Computing, Vienna, Austria).

## 3. Results

### 3.1. Study Participants

As a result of the measures to contain the COVID-19 pandemic in Switzerland during the entire project duration, recruiting schools and school classes for participating in the program proved very difficult. First, the containment measures led to a restriction of access of external persons, such as addiction prevention experts, to the school classes. Second, schools were organizationally strained during this period, and prevention services had low priority. Thirdly, the topic of alcohol consumption had less significance among young people during this period, due to the restrictions on going out and social life.

Nevertheless, the program could be implemented at a total of eight schools (seven upper secondary and one vocational school) in 61 school classes from February 2021 to July 2022. Figure 3 depicts participants’ progression through the trial. Within 49 school classes with an existing recruitment protocol, 965 students were present. Of these, 723 (59.4%) provided informed consent and registered for participation in the program and the associated study. Another 231 students were recruited from 12 school classes without a completed recruitment protocol, resulting in a total of 954 students from 61 school classes. Online follow up assessment at week 10 was completed by 272 students (28.4%).

The baseline characteristics for the study sample are presented in Table 2. The mean age of the program participants was 16.1 years (SD 1.0) and 56.6% of the participants were female.

Compared with non-participants, the program participants were more likely to be female (χ^2^ = 10.4, *p* < 0.01), younger (t = −3.3, *p* < 0.01), and hada lower self-efficacy to resist drinking (t = −3.3, *p* < 0.01) than the non-participants.

Online follow-up assessments were completed by 272 (28.4%) of the 954 program participants. Regarding attrition, the analysis revealed that follow-up assessments were completed more by female than male participants (χ^2^ = 14.2 *p* < 0.01), by students with higher self-efficacy to resist drinking alcohol (t = −3.4, *p* < 0.01), by those with a lower number of alcoholic drinks consumed per month (t = 2.8, *p* < 0.01), and by those with a lower number of alcoholic drinks consumed on one occasion (t = 2.4, *p* = 0.02). Regarding program use, it was found that those who participated in the follow-up assessment completed significantly more coaching dialogues within the program than those who did not (M = 8.6 vs. 1.8, t = −50.4, *p* < 0.01). To account for this attrition bias, these variables were entered as covariates within the models that evaluated changes in outcomes between baseline and 10-week follow-up.

### 3.2. Program Use

During the 10-week intervention period, participants were invited via weekly push notification to participate in 10 coaching dialogues. The mean number of completed weekly dialogues among the 954 program participants was 3.8 (SD = 3.6). No dialogue was completed by 13.2% (n = 126) and one dialogue was completed by 31.9% (n = 304). Between two and five dialogues were completed by 23.3% (n = 222). The proportion of users who completed at least six of the ten dialogues was 31.7% (n = 302).

In addition to the number of completed dialogues, the program usage time was calculated and operationalized as the time span between the first and last login to the program. This averaged 38.3 days (SD = 52.1 days). The proportion of users who used the program for at least half of the total program duration of 10 weeks, i.e., at least 5 weeks, was 44.7%.

### 3.3. Program Evaluation

Follow-up data on program evaluation were available from 240 (25.28%) of the 954 program participants. When evaluating the program overall, 84.6% found the tips and information helpful, 100.0% found them comprehensible, and 70.8% indicated that they felt they were relevant to their individual situation. Furthermore, 76.2% indicated they would recommend the program to others, 89.1% would participate again, and 87.9% said they enjoyed the program.

The “Ask-the-Expert” feature, the chatbot medium, and content, as well as the competition prizes, were rated as “very good” or “good” by around or over 90% of the respondent. The picture contests were rated as “good” or “very good” by two thirds of the participants (Figure 4).

### 3.4. Suggestions for Program Improvement

A total of 156 free-text responses were available to the questions “How could we improve the program overall?”, “Which topics or content should be given more attention?”, and “What additional media or functions would be useful?”.

The most frequent response was that the program was good as it was and that there was no need for improvement, with a total of 23 responses. Another 53 responses referred to a deepening or broadening of the range of topics. Among these, 16 people suggested that the topics of dependence, addiction, and the consequences of addiction should be discussed in greater detail. Eight people wished for an expansion with regard to other drugs such as nicotine or cannabis. Twelve statements referred to a more differentiated presentation of the medical and biological basis of addiction. In particular, they wished for more facts and statistics, as well as information, on the topics of “intoxication” and “hangover”. Six people wished for more information on how to deal with colleagues/family members who drink a lot of alcohol, three suggested more tips on how to deal with peer pressure/compulsion, and another three wished for an expansion of the range of topics in the direction of mental health (self-esteem and dealing with family conflicts). Two respondents each referred to a more differentiated presentation of the triggers for alcohol consumption and adolescent consumption patterns (irregular but high consumption). In addition, two persons wished for a stronger consideration of adult target groups, and one requested more tips for a safe consumption.

There were a total of 80 responses related to the technology, individual functions, and features of the program. For example, 22 respondents wished for greater individualization of the program, e.g., more answer options in the dialogs, the possibility of writing more free-text answers, and being able to freely choose the day and time of the coaching session. A total of 18 responses addressed the video clips within the program, of which seven wanted more videos, six wanted better videos, and five wanted shorter videos or a function to play the videos faster. Twelve people suggested integrating more interactive contests and challenges into the program, and five wanted more opportunities to interact with other program participants. Seven people wanted a consumption diary in which they could record their own alcohol consumption at shorter intervals. Three free text answers each suggested the integration of more quiz questions and the revision of the traffic light feedback. The elimination of technical difficulties when calling up links and uploading images was the topic of four responses. More frequent interaction with the chatbot was desired by three people. Another three people wanted fewer emojis in the dialogues with the chatbot.

### 3.5. Pre–Post Changes of Program Participants

#### 3.5.1. Binge Drinking

The GLMM analyses showed a statistically significant decline in the percentage of people with binge drinking in the previous 30 days between the baseline and follow-up assessment (OR 0.32, 95% CI 0.18–0.57, *p* < 0.01). Taking into account program participants with available follow-up data, the proportion of binge drinkers was 32.7% (89/272) at baseline and 24.3% (66/272) at follow-up.

#### 3.5.2. Maximum Alcohol Consumption

Regarding the maximum amount of drinks consumed on one occasion in the last 30 days, GLMM revealed a statistically significant decrease between the baseline and the follow-up survey (IRR 0.75, 95% CI 0.68–0.82, *p* < 0.01). Regarding program participants with available data at follow-up, the maximum amount of drinks consumed on an occasion was 3.2 (SD = 3.8) at baseline and 2.4 (SD = 3.0) at follow-up.

#### 3.5.3. Number of Standard Drinks per Month

According to the GLMM analyses, there was a statistically significant decrease in the total number of alcoholic standard drinks consumed in the previous 30 days between the baseline and the follow-up assessment (IRR 0.62, 95% CI 0.58–0.67, *p* < 0.01). Taking into account participants with available follow-up data, the average number of standard drinks decreased from 7.3 (SD = 13.9) at baseline to 4.5 (SD = 9.3) at follow-up.

#### 3.5.4. Drinking Refusal Self-Efficacy

Finally, the LMM analyses revealed a statistically significant increase in self-efficacy to resist drinking alcohol in different situations between the baseline and the follow-up assessment (Beta Coefficient 0.24, 95% CI 0.06–0.42, *p* = 0.01). Regarding program participants with available follow-up data, the mean number of standard drinks of program participants in a typical week was 4.8 (SD = 1.2) at baseline and 5.0 (SD = 1.0) at follow-up.

## 4. Discussion

### 4.1. Principal Results

The present study tested the acceptance and potential effectiveness of a newly developed mobile app-based coaching for alcohol prevention among adolescents. Three main findings were revealed: (1) Three of four students recruited in upper secondary and vocational school classes participated in program and the associated study, showing a high interest in the prevention program. (2) Based on program use and evaluations, the overall acceptance of the intervention was good. (3) The results concerning the initial effectiveness of this program derived from a pre–post investigation are promising.

Personal recruitment in school classes, in combination with offering a mobile app-based coaching program, allowed for reaching three out of four of the present students for participation in the *MobileCoach Alcohol* program and the associated study. This participation rate of 75% is comparatively high given the program’s 10-week duration, the requirement that users had to download a separate app on their smartphone, and the potentially lower interest in the topic of alcohol drinking during the COVID-19 pandemic and its related restrictions on going out and social life. Although no data are available for Switzerland to date, a recent review on substance use among young people during the COVID-19 pandemic suggests that the prevalence of youth alcohol use declined during the pandemic [23]. The participation rate of this new app-based coaching program is similar to that found in the RCT on the earlier web- and text-messaging-based program version within which 77% agreed to participate [12].

In comparison with other app-based prevention programs in school classrooms, program engagement in the *MobileCoach Alcohol* was good. For example, the mean number of dialogues processed in the *ready4life* prevention program [24], which is based on the same platform and addresses life skills in addition to addictive behaviors, was 2.1 out of 16 compared with 3.8 out of 10 in the *MobileCoach Alcohol*. Internationally, participant engagement and program use in app-based health interventions remain a challenge [25]. For example, a study of engagement in the app-based health promotion program “health4life” in Australia showed that of the 3610 learners invited to download the app in school classes, only 407 logged into the program at least once [26]. Among these, the average usage time was 10 days; for *MobileCoach Alcohol*, it was 38 days.

Despite the comparatively good program evaluations, there is still room for improvement with *MobileCoach Alcohol*. In particular, the evaluation of the interactive contests lagged behind the other program elements. The free text responses on program optimization show that there were also more frequent technical difficulties in the contests, e.g., when uploading images. Overall, among the suggestions for optimization, greater individualization of the coaching dialogs, an expansion to other substances and topics, and the communication of facts, especially on the topic of addiction and the effects of alcohol, were mentioned most frequently.

Although the results on the effectiveness of the program should be interpreted with caution due to the study design without a control group, the present data show comparable effect sizes as in the earlier program version as well as the controlled study on the previous *MobileCoach Alcohol* version [12]. In addition to the positive effects on drinking behavior, there was also a positive change in self-efficacy to resist drinking alcohol.

### 4.2. Limitations

The study’s main limitations include (1) a selective attrition with female participants, low-risk drinkers, and persons with high program use being more responsive. Therefore, a bias in the effectiveness and program evaluation results is possible. (2) The study was conducted in 2021/2022 during the COVID-19 pandemic, with varying restrictions that also influenced the social life and alcohol use of young people and accordingly the generalizability of the results of this study. (3) As we only included school classes that were willing to participate in the study, the results cannot be generalized to upper secondary and vocational school students in Switzerland.

## 5. Conclusions

The mobile app-based *MobileCoach Alcohol* program proved to be an attractive intervention, in which three of four students were interested when proactively recruited from school classes. Based on program use and evaluations, the overall acceptance of the intervention was good. It allows for individualized coaching in large groups of adolescents and young adults. Based on the presented initial results on the effectiveness, the program is promising for reducing at-risk alcohol use and increasing self-efficacy to resist alcohol. Based on these initial positive results, testing this interventional approach within a randomized controlled trial would be reasonable.

## Figures and Tables

**Figure 1 ijerph-20-03263-f001:**
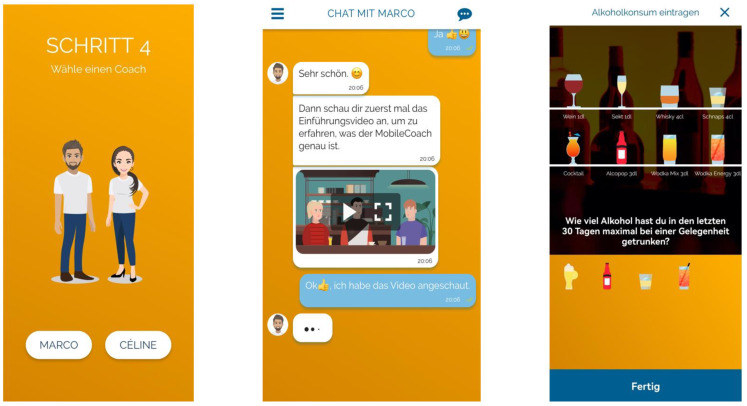
Screenshots from the baseline assessment of the MobileCoach Alcohol program (left: selection of coach; middle: introductory video; right: assessment of alcohol use).

**Figure 2 ijerph-20-03263-f002:**
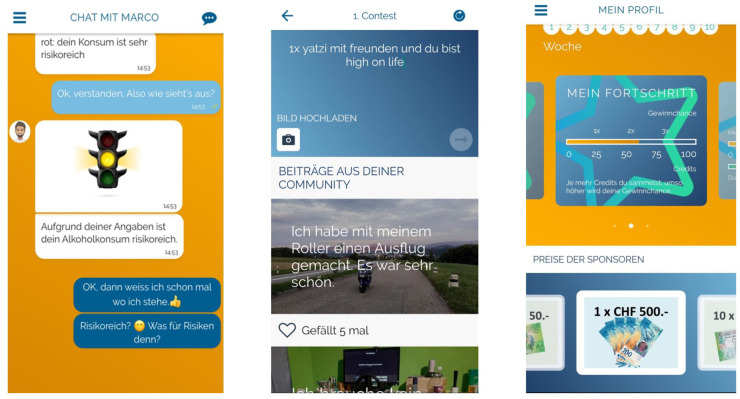
Screenshots from the *MobileCoach Alcohol* program (left: feedback on alcohol use; middle: contest on beautiful experience without alcohol; right: individual profile displaying progress within the program and potential prizes).

**Figure 3 ijerph-20-03263-f003:**
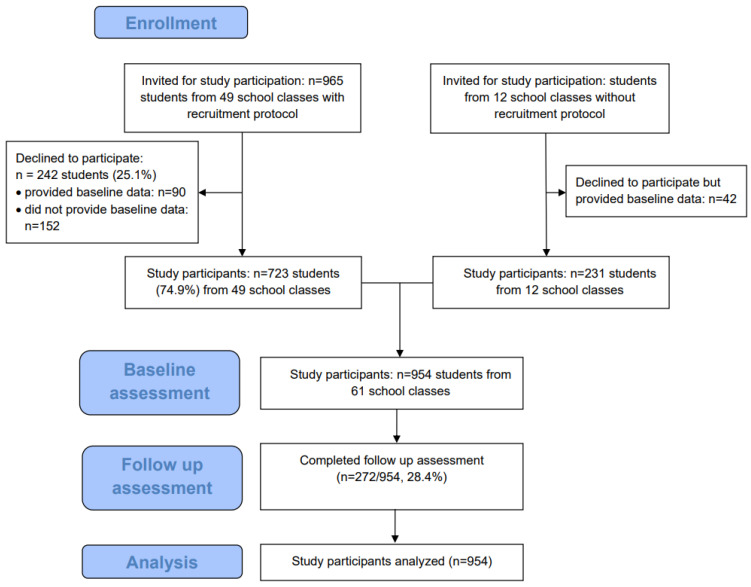
Flow chart of the study participants.

**Figure 4 ijerph-20-03263-f004:**
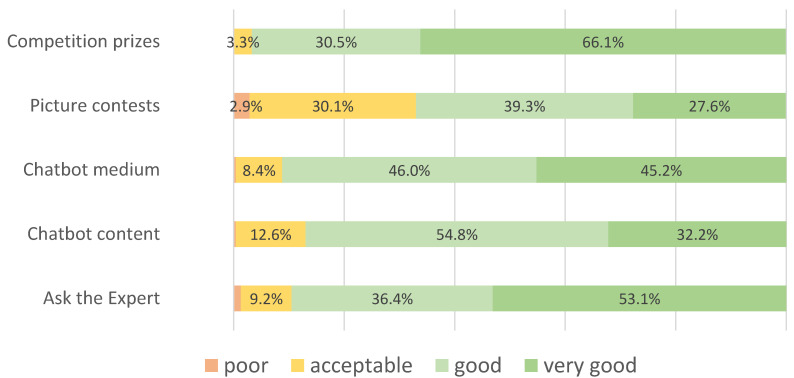
Evaluation of specific program elements of *MobileCoach Alcohol*. Values are presented for percentages >2%.

**Table 1 ijerph-20-03263-t001:** Content of the weekly coaching in the *MobileCoach Alcohol* program.

Week	Topic
1	Feedback on individual alcohol consumption, drinking recommendations
2	Video quiz on peer pressure
3	Pros and cons of alcohol consumption
4	Contest on beautiful experience without alcohol
5	Information on addiction in general
6	If-then plan for dealing with temptation situation
7	Ask-the-expert: Possibility to pose question to alcohol prevention expert
8	Communication challenge and quiz on role models
9	Video quiz on peer pressure, quiz on addictiveness of different substances
10	Frequently posed ask-the-expert questions and respective answers, follow-up survey and ipsative feedback on individual alcohol consumption
11	Reminder of follow-up survey

**Table 2 ijerph-20-03263-t002:** Baseline characteristics of the program participants and non-participants. Values represent n (%), unless stated otherwise.

Variable	Program Participants *n* = 954	Non-Participants *n* = 132	*p* ^a^
Sex			<0.01 ^b^
Male	414 (43.4%)	77 (58.3%)	
Female	540 (56.6%)	55 (41.7%)	
Age, mean (standard deviation)	16.1 (1.0)	16.5 (1.5)	<0.01 ^c^
Migration background No, both parents born in Switzerland Yes, at least one parent born abroad	530 (55.6%) 424 (44.4%)	66 (50.0%) 66 (50.0%)	0.23 ^b^
Binge drinking in previous 30 days			0.66 ^b^
No	430 (64.9%)	86 (65.2%)	
Yes	233 (35.1%)	46 (34.8%)	
Total number of alcoholic drinks consumed in the preceding 30 days, mean (standard deviation)	9.5 (17.8)	9.7 (19.7)	0.19 ^c^
Maximum number of alcoholic drinks consumed on one occasion in the preceding 30 days, mean (standard deviation)	3.7 (4.4)	3.5 (4.4)	0.73 ^c^
Self-efficacy to resist drinking alcohol, range 1–6 (low-high), mean (standard deviation)	4.6 (1.3)	5.0 (1.3)	<0.01 ^c^

^a^ *p* values for the comparison of the program participants and non-participants. ^b^ *χ^2^* test. ^c^ *t* test.

## Data Availability

Data are available upon request due to restrictions. The datasets generated and analyzed during the current study are not publicly available due to the Swiss data protection law, but are available from the corresponding author upon reasonable request. Requests will be reviewed for reasonability and compliance with the study purpose and the participants’ informed consent.

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
