# Peer review of "Mobile App-Based Coaching for Alcohol Prevention among Adolescents: Pre–Post Study on the Acceptance and Effectiveness of the Program “MobileCoach Alcohol”"

_ijerph, 2023, doi:10.3390/ijerph20043263_

Round 1
Reviewer 1 Report
The authors drew attention to a significant problem among adolescents and young adults, which is alcohol consumption. For this purpose, they tried to assess the usefulness of the mobile application among vocational and upper secondary school students in Switzerland. Despite some issues, this study is interesting and offers new information.
1. Introduction:
It should be emphasized that not only long-term alcohol consumption is a risk of liver disease, but even a single use of alcohol has a negative impact on its functioning. DOI: 10.3390/biom11060911
2. Methods:
The authors list only the criteria for inclusion of participants in the study. What were the exclusion criteria?
I propose to describe the role of the coach in more detail. Did each participant receive the same messages? Were they personalized?
Why are only 3 demographic factors (age, sex, migration background) taken into account? Perhaps it is worth taking into account the influence of other socio-demographic factors on the obtained results and describing it in the discussion.
The T-student test was used in the statistical analysis. What normality test was used to check this?
3. Results:
The abbreviation "M" is used in Table 1. Its meaning should be explained.
Is a value of 1 on the "self-efficacy to resist drinking alcohol" scale the lowest or highest value?
The terms "corona virus" or "corona" are not used in available medical publications.
Authors should consider including more results in tables to make the data more clear and transparent.
4. Discussion:
Have data on alcohol consumption by adolescents during the COVID-19 pandemic been published? It is worth taking this into account in the discussion, since the research took place during the pandemic restrictions.
Author Response
Reviewer 1
The authors drew attention to a significant problem among adolescents and young adults, which is alcohol consumption. For this purpose, they tried to assess the usefulness of the mobile application among vocational and upper secondary school students in Switzerland. Despite some issues, this study is interesting and offers new information.
- Introduction:
It should be emphasized that not only long-term alcohol consumption is a risk of liver disease, but even a single use of alcohol has a negative impact on its functioning. DOI: 10.3390/biom11060911
Reply
Thanks for this information. We included this and your reference in the corresponding sentence on page 2, 2nd paragraph: "At-risk alcohol use among adolescents and young adults is associated with social and educational problems, accidents, damage to the liver, and, in the longer term, increased risk of chronic diseases, such as heart and liver disease or alcohol dependence (Kuntsche, Kuntsche, Thrul, & Gmel, 2017; Zdanowicz et al., 2021).
- Methods:
The authors list only the criteria for inclusion of participants in the study. What were the exclusion criteria?
Reply
We have not defined any fixed exclusion criteria. In principle, the students in the school classes had sufficient language skills and cognitive abilities and were eligible for participation if they had a smartphone and were of minimum age.
I propose to describe the role of the coach in more detail. Did each participant receive the same messages? Were they personalized?
Reply
Thanks, we described the role of the coach in more detail on page 4 (1st paragraph).
"The coach regularly contacted the participants via push notifications and guided them through the program. The coaching content was individualized based on age, sex, and alcohol consumption (at-risk or not at risk, days of consumption, times of consumption, consumption situations)."
Why are only 3 demographic factors (age, sex, migration background) taken into account? Perhaps it is worth taking into account the influence of other socio-demographic factors on the obtained results and describing it in the discussion.
Reply
Although other variables such as health literacy or socioeconomic status would have been interesting from a research point of view, we had to limit ourselves to a minimum in order to reduce the proportion of dropouts in the initial survey within the app.
The T-student test was used in the statistical analysis. What normality test was used to check this?
Reply
We used the Kolmogorov-Smirnov Test and a visual inspection of the distribution of the data to check the normal distribution assumption.
- Results:
The abbreviation "M" is used in Table 1. Its meaning should be explained.
Is a value of 1 on the "self-efficacy to resist drinking alcohol" scale the lowest or highest value?
The terms "corona virus" or "corona" are not used in available medical publications.
Authors should consider including more results in tables to make the data more clear and transparent.
Reply
Thanks, we have written out and added the terms mean and standard deviation in Table 2 on page 8.
Furthermore, we have added in this Table that a value of 1 corresponds to low self-efficacy and 6 to high self-efficacy.
The terms "corona virus" and "corona" were replaced by "COVID-19" throughout the manuscript.
- Discussion:
Have data on alcohol consumption by adolescents during the COVID-19 pandemic been published? It is worth taking this into account in the discussion, since the research took place during the pandemic restrictions.
Reply
Thanks, we added this in the discussion on page 11:
"Although no data are available for Switzerland to date, a recent review on substance use among young people during the COVID‑19 pandemic suggests that the prevalence of youth alcohol use has declined during the pandemic (Layman et al., 2022)."
Reviewer 2 Report
The text deals with an essential topic of alcohol addiction among adolescents and potential prevention through a mobile application. The authors prove that the app they developed has some results among young people in terms of lowering the statistics of binge drinking, the maximum amount of alcohol consumed, the monthly number of drinks, and increasing the effectiveness in self-efficacy to resist drinking alcohol.
Among the data collected at the outset, we have the question of migratory background, while we do not have the question of the family situation (single parent, both parents, no parents), which generally has a significant impact on the youth's self-understanding.
The researchers said that Covid made it difficult for them to conduct the study properly because of the restrictions, the youths had fewer encounters where there would be opportunities to consume alcohol. Other studies indicate that due to the solitude during covid, they were more susceptible to addiction and mental disorders.
While the application seems useful, the question arises as to how it relates to the Kantian principle of autonomy. The dignity of man requires, according to Kant, that he in his interior make conscious and voluntary decisions. It is opposed to being subject to external control. It seems that the app could be a tool for such external control of young people's attitudes and behavior. Especially that contact with a living person in time appears in it sporadically. Researchers are conscious of this fact placing media extensive use on the list of addictions.
The app itself is likely to be marginally addictive but may contribute to the vulnerability of young people to such addiction. Preventing one addiction would then pave the way to another addiction. It seems it would be useful to have a few sentences about this in the introduction or conclusions, and some warning in the app itself.
Program participants highly valued the opportunity to win the prize. The question arises whether it is appropriate to build such motivation for abstinence.
Author Response
Reviewer 2
The text deals with an essential topic of alcohol addiction among adolescents and potential prevention through a mobile application. The authors prove that the app they developed has some results among young people in terms of lowering the statistics of binge drinking, the maximum amount of alcohol consumed, the monthly number of drinks, and increasing the effectiveness in self-efficacy to resist drinking alcohol.
Among the data collected at the outset, we have the question of migratory background, while we do not have the question of the family situation (single parent, both parents, no parents), which generally has a significant impact on the youth's self-understanding.
Reply
Although other variables such as family situation, health literacy or socioeconomic status would have been interesting from a research point of view, we had to limit ourselves to a minimum in order to reduce the proportion of dropouts in the initial survey within the app.
The researchers said that Covid made it difficult for them to conduct the study properly because of the restrictions, the youths had fewer encounters where there would be opportunities to consume alcohol. Other studies indicate that due to the solitude during covid, they were more susceptible to addiction and mental disorders.
Reply
Thanks for this comment, we added a review on alcohol use during the COVID-19 pandemic in the discussion section:
"Although no data are available for Switzerland to date, a recent review on substance use among young people during the COVID‑19 pandemic suggests that the prevalence of youth alcohol use has declined during the pandemic (Layman et al., 2022)."
While the application seems useful, the question arises as to how it relates to the Kantian principle of autonomy. The dignity of man requires, according to Kant, that he in his interior make conscious and voluntary decisions. It is opposed to being subject to external control. It seems that the app could be a tool for such external control of young people's attitudes and behavior. Especially that contact with a living person in time appears in it sporadically. Researchers are conscious of this fact placing media extensive use on the list of addictions.
The app itself is likely to be marginally addictive but may contribute to the vulnerability of young people to such addiction. Preventing one addiction would then pave the way to another addiction. It seems it would be useful to have a few sentences about this in the introduction or conclusions, and some warning in the app itself.
Reply
Thank you for these critical comments. As you rightly point out, there are also behavioural addictions, such as gambling or gaming. Extensive media use is not yet recognised as an addiction in either the DSM or the ICD. However, because the app provides for a limited number of interactions (10 weeks, one to three weekly dialogues with the eCoach), there is no danger of excessive use. The users have the option to end their participation at any time, both when they are invited to participate and during the use of the program. This is also explicitly emphasised in the program information and was approved by the ethics committee.
Although I and other addiction researchers consider certain online activities such as excessive social media or pornography use to be critical and call for more research on these issues, general media use is not considered critical. Media such as newspapers, books, TV and the internet are part of our everyday lives and as such are not a cause for concern.
Program participants highly valued he opportunity to win the prize. The question arises whether it is appropriate to build such motivation for abstinence.
Reply
Thanks for this comment. We have deliberately refrained from using the competition prizes to achieve a behavioral goal such as abstinence or alcohol consumption reduction. The chance to win a prize was only linked to participation in the program. The motivation to use the app regularly should be increased by the competition prizes.
Reviewer 3 Report
Introduction: 5 th para 56 th line mHealth should be replaced with health. Figure 1. depicts the mobile app where language is swiss may be, however for better understanding to mass people, English would be better I believe. Conclusion text must be increased with key findings.
Author Response
Reviewer 3
Introduction: 5 th para 56 th line mHealth should be replaced with health. Figure 1. depicts the mobile app where language is swiss may be, however for better understanding to mass people, English would be better I believe. Conclusion text must be increased with key findings.
Reply
Thanks, we replaced "mHealth interventions" by the more comprehensive term "mobile phone-based interventions" on page 2, 5th paragraph.
Unfortunately, the app and the screenshots of the app are not available in English. However, we believe that the figure captions provide a rough understanding of the contents. If this is not sufficient from your point of view, a translation in the captions would be possible.
The key findings were added to the conclusions text which now reads as follows:
"The mobile app-based MobileCoach Alcohol program proved to be an attractive intervention, in which three of four students are interested when proactively recruited at school classes. Based on program use and evaluations, the overall acceptance of the intervention was good. It allows an individualized coaching in large groups of adolescents and young adults. Based on the presented initial results on effectiveness, the program is promising for reducing at-risk alcohol use and increasing self-efficacy to resist alcohol. Based on these initial positive results, testing this interventional approach within a randomized controlled trial would be reasonable."